# Point-of-care HIV viral load testing in a community antiretroviral therapy delivery programme: A randomised controlled trial (PHILA)

Jienchi Dorward[1,2]*, Kwena Tlhaku[2,3], Yukteshwar Sookrajh[4], Pedzisai Munatsi[2], Jessica Naidoo[2], Emelda Tselana[2], Andile Maphumulo[2], Nokuthandwa Mbambo[2], Thobile Mhlongo-Gumbi[2], Jennifer A. Brown[1,2], Lara Lewis[2], Pravikrishnen Moodley[5], Natasha Samsunder[2], Paul K. Drain[6,7,8], Christopher C. Butler[1], Gail Hayward[1], Nigel Garrett[2,3,9]

1 Nuffield Department of Primary Care Health Sciences, University of Oxford, Oxford, United Kingdom, 2 Centre for the AIDS Programme of Research in South Africa (CAPRISA), University of KwaZulu–Natal, Durban, South Africa, 3 Discipline of Public Health Medicine, School of Nursing and Public Health, University of KwaZulu-Natal, Durban, South Africa, 4 eThekwini Municipality Health Unit, eThekwini Municipality, Durban, South Africa, 5 Department of Virology, University of KwaZulu-Natal and National Health Laboratory Service, Inkosi Albert Luthuli Central Hospital, KwaZulu-Natal, South Africa, 6 Department of Global Health, Schools of Medicine and Public Health, University of Washington, Seattle, Washington, United States of America, 7 Department of Medicine, School of Medicine, University of Washington, Seattle, Washington, United States of America, 8 Department of Epidemiology, School of Public Health, University of Washington, Seattle, Washington, United States of America, 9 Desmond Tutu HIV Centre, University of Cape Town, Cape Town, South Africa

* jienchi.dorward@phc.ox.ac.uk

## Abstract

Community antiretroviral therapy (ART) delivery programmes allow people with HIV (PWH) to collect treatment nearer to home instead of from clinics. However, delays in laboratory-based viral load (VL) testing can prevent timely community ART prescription renewals. We aimed to determine if clinic point-of-care VL testing could expedite community ART prescription renewals within the Centralised Chronic Medication Dispensing and Distribution programme (CCMDD) in South Africa. We conducted an open-label, randomised controlled trial of point-of-care versus laboratory-based VL testing among PWH who needed community ART CCMDD prescription renewal in one clinic in Durban, South Africa. The primary outcome was community ART CCMDD prescription renewal by three weeks. We enrolled 200 participants between August, 2022 and August, 2023. Median age was 44 years (interquartile range [IQR] 37–49), and 65.5% were female. 93/100 (93.0%) intervention arm participants had a community ART CCMDD prescription renewal within three weeks, versus 81/100 (81.0%) standard-of-care participants (risk difference [RD] 12.0%, 95% confidence interval [CI] 2.9 to 21.2%, p = 0.021). Participants received VL results after a median 0 days (IQR 0–0) in the intervention and 20 days (IQR 7 to not received) in the

**Data availability statement:** Data cannot be shared publicly because of participant confidentiality considerations and our institutional responsibility to ensure that any data we collect is used ethically by other researchers in order to maintain public confidence and trust in the research we conduct. All requests for data access are reviewed by Nuffield Department of Primary Care's Primary Care Hosted Research Datasets Independent Scientific Committee (PrimDISC). PrimDISC ensures that data collected in the public interest is used responsibly, ethically and not for spurious reasons. Requests for data access will not be unreasonably refused. Researchers wishing to access data should contact primdisc@phc.ox.ac.uk.

**Funding:** This work was supported by grants from the Dowager Countess Eleanor Peel Trust (#280 to JD and NG), the Wellcome Trust PhD Programme for Primary Care Clinicians (216421/Z/19/Z to JD), the Tropical Health Education Trust (no grant number, to JD), and the Gates Foundation ([INV-051067 to JD and NG]). The conclusions and opinions expressed in this work are those of the authors alone and shall not be attributed to the Foundation. For the purpose of open access, and under the grant conditions of the Foundation, the author has applied a CC BY public copyright licence to the Author Accepted Manuscript version that might arise from this submission. JAB is funded by the Swiss National Science Foundation (P500PM-221966). JD, Academic Clinical Lecturer (CL-2022-13-005), is funded by the UK National Institute of Health and Social Care Research (NIHR). The views expressed are those of the authors and not necessarily those of the NHS, the NIHR or the Department of Health and Social Care. The funders had no role in study design, data collection and analysis, decision to publish, or preparation of the manuscript.

**Competing interests:** I have read the journal's policy and the authors of this manuscript have the following competing interests: Cepheid provided point-of-care viral load assays at no cost for use at the study site. The authors have declared that no other competing interests exist.

standard-of-care arm. There was no difference between arms in the proportion retained-in-care between 8 and 16 weeks (89.0% versus 87.0%, RD 2.0% 95% CI -8.0 to 12.0). For community ART CCMDD prescription renewal the mean number of clinic visits required was lower in the intervention arm (1.06) versus the standard-of-care arm (1.60, RD -0.54, 95% CI -0.40 to -0.68), as was the total participant travel cost to participants (South African Rands [ZAR] 47.7 versus ZAR 72.8, RD ZAR -25.1 [95% CI -9.2 to -41.1]). Point-of-care VL testing improved community ART prescription renewals, by reducing time to results, and reducing the number of clinic visits and associated travel costs. Pan-African Clinical Trials Registry: PACTR202002785960123.

## Introduction

South Africa has the largest antiretroviral therapy (ART) programme globally, with over 5 million people receiving ART. In order to provide chronic medication efficiently, the South African National Department of Health (SANDoH) introduced the Centralised Chronic Medication Dispensing and Distribution system (CCMDD) [1], which provides ART to over 2 million people [2]. CCMDD can benefit people living with HIV (PLWH) by facilitating ART collection at community-based pick-up points (churches, community groups, pharmacies) that are nearer, have longer opening hours and are less congested than clinics [3,4]. However, barriers to implementation of CCMDD [3,5] have resulted in between 16–26% of CCMDD participants being 'dormant' [6], meaning their CCMDD prescriptions have lapsed by more than three weeks.

Reducing the number of dormant clients has been a SANDoH priority, but has proved difficult. One potential cause is the blood tests required to determine CCMDD eligibility, as only people with a suppressed HIV viral load (VL) <50 copies/mL are eligible for enrolment into CCMDD, and prior to 2023, South African guidelines required a confirmed annual VL < 1000 copies/mL for clients to have their CCMDD prescription renewed [7]. VLs are a potential health systems barrier to enrolment and continuation in CCMDD, as testing is conducted at centralized laboratories, meaning clients generally have to attend a clinic twice for a blood draw and then review of results, with results sometimes being delayed or lost (Fig A in S1 Text) [8]. This can lead to delays in renewal of CCMDD prescriptions, and increased numbers of dormant clients continuing to attend clinics rather than collecting ART in the community.

Point-of-care VL testing may help mitigate this problem and reduce the number of dormant clients, by providing same day results that allow CCMDD referral in one clinic visit [9]. However, whether point-of-care VL testing can be successfully integrated into CCMDD clinical pathways, with results used by routine, front-line healthcare workers is not known. Therefore, we aimed to assess whether point-of-care VL testing could increase CCMDD renewals and reduce time to ART collection in CCMDD in a public clinic in South Africa.

## Methods and analysis

### Ethics statement

The University of KwaZulu-Natal Biomedical Research Ethics Committee (BREC/00000837/2019 and BE646/17) and the University of Oxford Tropical Research Ethics Committee (OxTREC 64–19) approved the Point-of-care HIV viral Load testing in a community Antiretroviral therapy delivery programme (PHILA) study, which was registered on the Pan African Clinical Trials Registry (PACTR202002785960123, https://pactr.samrc.ac.za/TrialDisplay.aspx?TrialID=9727) on 12th February 2020. The University of Oxford is the study sponsor. Formal written informed consent was obtained from all PHILA participants.

### Trial design

The PHILA study, was a single-site, open-label, individually randomised trial of point-of-care HIV VL testing among PLWH in the CCMDD programme. The full protocol is available online [10].

The study was conducted at the Prince Cyril Zulu Communicable Disease Centre (PCZ CDC), a large, public clinic next to the central Durban transport hub, with support from the adjacent Centre for the AIDS Programme of Research in South Africa (CAPRISA) eThekwini Clinical Research Site. PCZ CDC provides HIV, tuberculosis and other primary care services and is in the province of KwaZulu-Natal, which has an estimated HIV prevalence of 27% among adults aged 15–49 [11]. HIV treatment is provided free at the point of service, according to South African Department of Health guidelines [7], which at the time of the study recommended HIV VL testing at six months after ART initiation, and then annually. People were eligible for CCMDD if they were clinically stable with a VL < 50 copies/ml and estimated glomerular filtration rate (eGFR) >50 mL/min/1.73 m$^2$ in the past 12 months, not pregnant, and did not have any medical condition requiring frequent clinical monitoring such as tuberculosis or uncontrolled diabetes or hypertension. Eligible people could be referred by a clinic nurse, who would prescribe six months of ART, with the first two or three months provided at the referral clinic visit. After two or three months, the client would then collect a further three-month supply of ART, or two further two-month supplies, at the pick-up point of their choice, before the client would return to the clinic for assessment and a new prescription after six months (Fig A in S1 Text). Every 12 months, an annual viral load would be taken, either a few days earlier than the CCMDD prescription renewal was due (so that the result was available on the scheduled renewal date), or on the date the renewal was due, with one month supply of ART provided, and the result assessed at a follow-up visit after 7–28 days (Fig A in S1 Text). People with a VL ≥ 1000 copies/mL would be de-registered from CCMDD, while those with a VL between 50–999 copies/mL could continue in CCMDD, but would require enhanced adherence counselling at the clinic. Renewed CCMDD prescriptions would again be for six months of ART, with the first two or three months provided at the referral clinic visit, and subsequent ART collected from pick-up points (Fig A in S1 Text) including private pharmacies, community group venues and smart lockers known as Pele-Boxes, from which patients can retrieve pre-packed medication using a pin code sent to their phone [12].

### Participants

People were eligible for enrolment into PHILA if they were living with HIV, aged ≥18 years, receiving ART in CCMDD and due a VL test at the clinic for a renewal of their CCMDD prescription. People who were known to be pregnant or breast feeding, had current tuberculosis, known diabetes with blood glucose >7.0mmol/L, known hypertension with blood pressure ≥140/90 mmHg, or other medical conditions requiring regular clinical consultations, were ineligible for CCMDD and PHILA.

### Enrolment and randomisation

After providing written informed consent and screening, eligible participants were enrolled and randomised by a study nurse, using a pre-programmed REDCap REDCap (Research Electronic Data Capture) [13], electronic case report form,

in a 1:1 ratio to the point-of-care arm or the standard-of-care arm. A statistician generated the allocation sequence using computer generated random numbers with variable block sizes. All other study staff were blinded to the block sizes and allocation sequence. Clinic staff and participants were told the participant's allocation at enrolment.

### Interventions

At enrolment and during follow up, clinical management was provided by public-sector clinic counsellors, nurses or clinicians, according to South African Department of Health guidelines [7]. VL and creatinine testing was conducted according to the study arm.

**Point-of-care testing.**  For participants randomised to the point-of-care arm, creatinine testing was conducted by a research nurse from a capillary finger-prick sample using the Statsensor Xpress-I (Nova Biomedical, Waltham, USA), which provides a result in 90 seconds. VL testing was conducted using the Xpert HIV-1 VL XC assay (Cepheid, Sunnyvale, USA), which measures quantitative VL in 90 minutes. We planned for clinic nurses to do the point-of-care VL testing in the study clinic, but staff shortages and COVID-19 related disruption meant that it was initially conducted on a 4 module GeneXpert machine by a research laboratory technician at the clinic site laboratory. Later, research nurse testing in a clinical room using two single module GeneXpert machines was introduced, with support from laboratory technicians. Both laboratory technicians and nurses received on-site manufacturer training and ongoing support to conduct testing according to manufacturer instructions. In brief, a venous blood sample was centrifuged to provide 1ml of plasma, which was tested using the Xpert HIV-1 VL XC cartridge. For invalid results, leftover plasma was used for retesting, or a repeat sample was requested. Participants were encouraged to wait for results (Fig A in S1 Text), but if they were unwilling, results were provided at their next clinic appointment, scheduled by clinic staff in consultation with participants at the soonest possible date. Once point-of-care results were available, they were provided to routine clinic staff. Those with a $VL < 50$ copies/mL and $eGFR > 50$ mL/min/1.73 $m^2$ were referred to CCMDD, while those with a $VL \geq 50$ copies/mL or $eGFR \leq 50$ mL/min/1.73 $m^2$ were referred to a counsellor or clinician for further management.

**Laboratory-based VL testing.**  For participants randomised to the standard-of-care arm, venous blood was drawn by a research nurse and sent for creatinine and VL testing off-site by the National Health Laboratory Service. VL testing was conducted using the Alinity m HIV-1 VL analyser (Abbott, Chicago, USA). Initially, participants would return to the clinic at a date arranged at their and the healthcare workers convenience, but normally after 7–28 days, to be assessed for CCMDD eligibility based on VL and creatinine results (Fig A in S1 Text). However, almost exactly halfway through enrolment, on 24th May 2023, new South African guidelines were implemented which recommended that in the standard-of-care arm, clients could have their CCMDD prescription renewed without waiting for the laboratory results (Fig A in S1 Text), and that they would only be called back if the VL was $\geq 50$ copies/mL or $eGFR \leq 50$ mL/min/1.73 $m^2$ [14]. Otherwise, they would receive the results when they returned to clinic six months later.

### Follow-up

Participants were not followed-up in-person by research staff. Instead, outcomes were ascertained up to 16 weeks using reviews of participants' routine clinical charts, laboratory results and the CCMDD electronic database. If by 16–18 weeks there was no evidence of CCMDD renewal prescription or ART collection, the research team attempted to contact the participant to establish the reason for not collecting ART in CCMDD.

### Outcomes

The primary outcome was renewal of CCMDD prescription within three weeks of enrolment. We chose this outcome to align with the programmatic definition of being dormant if a CCMDD prescription had lapsed by three weeks. Secondary outcomes were time (days) from enrolment to CCMDD prescription renewal, days to first CCMDD ART collection, retention-in-care (defined as documented ART collection between 6–16 weeks post-enrolment), days from enrolment until

participants received their VL results, number of clinic visits from enrolment until CCMDD prescription, and travel costs in South African Rands (ZAR, 1 ZAR ~ 18 United States Dollars) from the patient perspective to have their CCMDD prescription renewed. As PHILA did not involve an investigational medicinal product or intervention that affects physiology, only adverse events of hospitalisations and deaths were recorded [10].

## Sample size and statistical methods

Assuming that 75% of participants in the standard-of-care arm would achieve the primary outcome of CCMDD renewal by 3 weeks, and a 15% improvement in the point-of-care arm, we calculated that a sample size of 100 participants per arm would give us 80% power to demonstrate superiority with a two-sided alpha of 0.05% [15,16].

In pre-specified analyses, we calculated the proportions of participants achieving study outcomes and compared proportions in each arm using the chi-squared test, and presented absolute risk differences in percentage points. We included all participants enrolled in each arm in intention-to-treat analyses. For secondary time-to-event outcomes, we did not use Cox proportional hazards models due to the high proportion of tied events in the point-of-care arm on days 0 and 1, and the violation of the proportional hazards assumption. Instead, as pre-specified in the protocol, we compared proportions achieving the outcome of interest at 2, 4, 8, 12 and 16 weeks post enrolment, using chi-squared tests and Newcombe Wilson 95% confidence intervals [17]. We also drew Kaplan Meier survival curves. Amongst those with CCMDD renewals, we compared the mean number of clinic visits from enrolment until CCMDD prescription renewal, and the mean total travel cost to the participant to attend these clinic visits (number of visits x travel cost per visit), using Students t-Test. In a post-hoc analysis, we used a Poisson regression model with robust standard errors [18] and an interaction term between time period and study arm, to evaluate whether there was a sub-group effect depending on enrolment before or after the guideline changed to allow same-day CCMDD renewal in the standard-of-care arm without receipt of VL results.

Data was collected and managed using REDCap [13], and analysed using R v4.1 [19].

## Post hoc evaluation of the impact of same day laboratory viral load testing and CCMDD prescription renewal in public clinics using routine de-identified data

Lastly, we evaluated the broader impact, outside the PHILA trial, of the guideline change to allow clients to have their CCMDD prescription renewed on the same day as the laboratory VL test, without waiting for the results. Specifically, we aimed to assess whether this resulted in people being referred to CCMDD while still viraemic, and whether those with viraemia were managed according to the guidelines which recommend prompt recall to the clinic for enhanced adherence counselling and a repeat viral load after 3 months. We analysed de-identified, routinely collected TIER.Net [20] CCMDD referral and viral load data from 108 primary care clinics in KwaZulu-Natal, from 24th May (date of guideline change) to 10th September (16 weeks before data cut of 31st December), 2023. We evaluated 1) the number of non-pregnant adults ≥18 years old in CCMDD who had their community ART CCMDD prescription renewed on the same day as the laboratory VL was taken, 2) the proportion of VLs with viraemia >50 copies/mL and ≥1000 copies/mL, and of these, the proportion who had 3) a clinic visit and 4) a repeat VL within 16 weeks (corresponding to the PHILA follow-up time and the recommendation to have a repeat VL within 3 months).

## Results

### Study population

Between August 15th, 2022 and August 24th, 2023, we assessed 899 people in CCMDD for eligibility, and enrolled 200 participants (Fig 1). Median age was 44 years (interquartile range [IQR] 37–49), 65.5% were female, and median time on ART was 8.0 years (IQR 6.0-10.9) (Table 1). 83.0% were receiving tenofovir disoproxil fumarate, lamivudine and dolutegravir (TLD), and 14.5% tenofovir disoproxil fumarate, emtricitabine and efavirenz (TEE). Median time since the participant had been first referred to CCMDD was 3.8 years (IQR 1.8 to 6.0), and median time since the last CCMDD referral

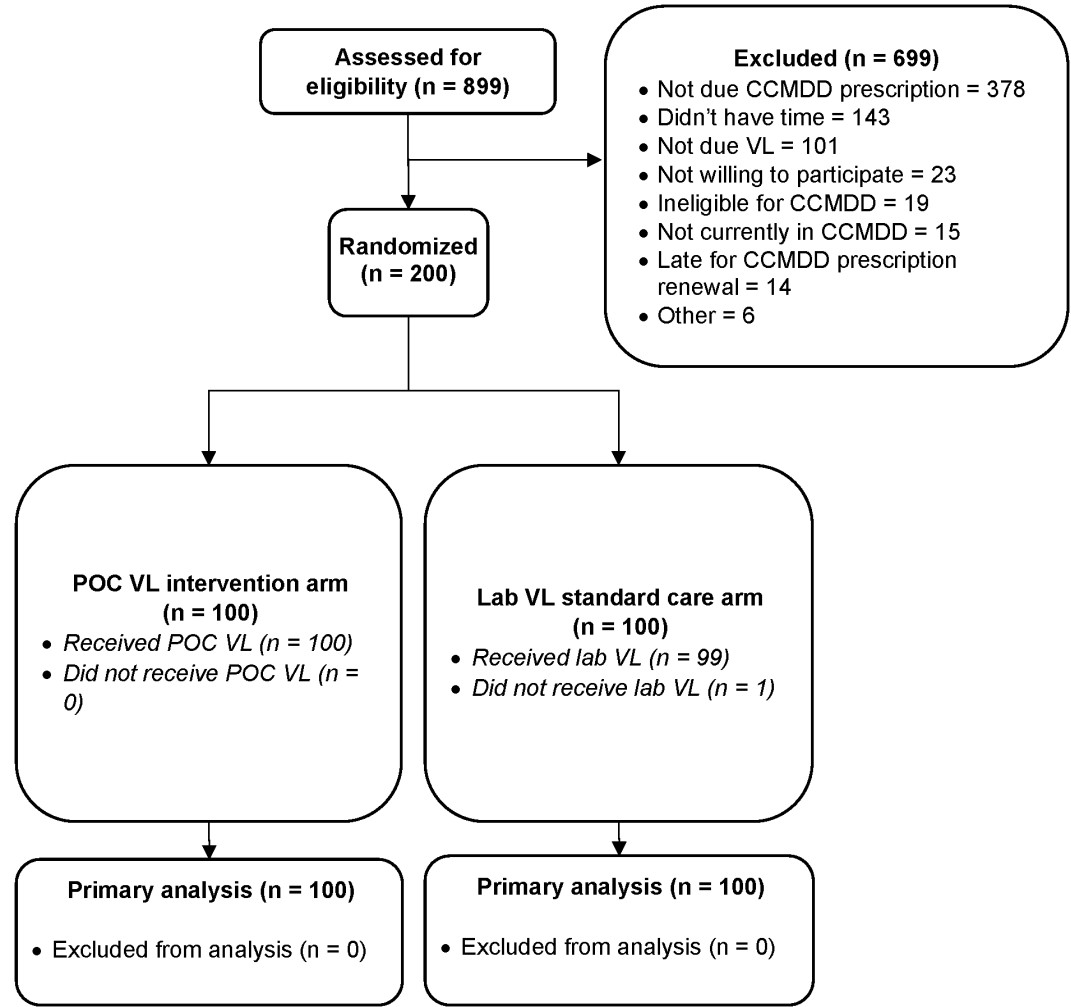

**Fig 1. PHILA study CONSORT diagram.** POC = point-of-care, VL = viral load, EAC = enhanced adherence counselling.

was 168 days (IQR 161–172). Median monthly income was ZAR 3500 (IQR 350–5000). The median time and cost to travel to the clinic and back was 60 minutes (IQR 50–90) and ZAR 44 (28–50) respectively. The median time and cost to travel to the last used pick-up-point and back was 50 minutes (IQR 30–60) and ZAR 30 (IQR 20–50) respectively.

There were slightly higher proportions of women (70% versus 61%) and participants reporting no monthly income (28% versus 19%) in the intervention arm. There were more participants reporting hazardous alcohol use (76% versus 66%) in the standard-of-care arm, and the median CD4 count at initiation (308 versus 226 cells/μL) was higher in the standard-of-care arm. Other demographics and clinic variables were well balanced between the two groups.

Enrolment VLs were suppressed <50 copies/mL for 94% of the intervention arm and 92% of the standard-of-care arm (Table 2). In the intervention arm, site laboratory staff conducted 74 (74%) and nurses 26 (26%) of the point-of-care VL tests. There were four invalid enrolment point-of-care results, which were repeated and results given to participants on the same day; all were <50 copies/mL. There were three invalid enrolment laboratory VL results, which were subsequently repeated after 5, 8 and 45 days from enrolment, with results reaching the patient 99, 11 and 48 days from enrolment respectively.

**Table 1. Baseline characteristics of PHILA study participants, n = 200.**

| Variable | Levels | Intervention arm | Standard-of-care arm | Total |
|---|---|---|---|---|
| **Demographics** | | | | |
| Age, years | Median (IQR) | 44.0 (37.8 to 49.0) | 44.0 (36.8 to 49.0) | 44.0 (37.0 to 49.0) |
| Gender | Female | 70 (70.0) | 61 (61.0) | 131 (65.5) |
| | Male | 30 (30.0) | 39 (39.0) | 69 (34.5) |
| Ethnicity | Black African | 99 (99.0) | 100 (100.0) | 199 (99.5) |
| | Other | 1 (1.0) | | 1 (0.5) |
| **Clinical information** | | | | |
| Time since HIV diagnosis, years | Median (IQR) | 9.3 (7.0 to 12.0) | 8.9 (6.5 to 12.1) | 9.1 (6.7 to 12.0) |
| Time since ART initiation, years | Median (IQR) | 8.9 (6.7 to 10.9) | 7.9 (6.0 to 10.2) | 8.0 (6.0 to 10.9) |
| Initiation CD4 count, cells/µL | Median (IQR) | 225.5 (140.0 to 339.0) | 308.0 (196.5 to 456.5) | 251.0 (158.5 to 387.5) |
| Initiation CD4 count category, cells/µL | <200 | 40 (40.0) | 23 (23.0) | 63 (31.5) |
| | 200-349 | 29 (29.0) | 27 (27.0) | 56 (28.0) |
| | 350-499 | 11 (11.0) | 21 (21.0) | 32 (16.0) |
| | >=500 | 8 (8.0) | 16 (16.0) | 24 (12.0) |
| | (Missing) | 12 (12.0) | 13 (13.0) | 25 (12.5) |
| Current ART regimen at enrolment | TDF/ 3TC/ DTG | 86 (86.0) | 80 (80.0) | 166 (83.0) |
| | TDF/ FTC/ EFV | 14 (14.0) | 15 (15.0) | 29 (14.5) |
| | Other | | 5 (5.0) | 5 (2.5) |
| Time on current regimen, years | Median (IQR) | 2.7 (1.8 to 2.9) | 2.7 (1.9 to 3.0) | 2.7 (1.8 to 2.9) |
| Time since first CCMDD referral, years | Median (IQR) | 4.0 (1.8 to 6.3) | 3.7 (1.8 to 5.9) | 3.8 (1.8 to 6.0) |
| Time since latest CCMDD referral, days | Median (IQR) | 168.0 (164.0 to 168.2) | 168.0 (161.0 to 174.2) | 168.0 (161.0 to 172.2) |
| Time since pre-enrolment viral load, days | Median (IQR) | 358.5 (342.5 to 371.0) | 356.5 (349.0 to 371.0) | 357.0 (344.0 to 371.0) |
| Pre-enrolment viral load category, copies/mL | < 50 | 98 (98.0) | 100 (100.0) | 198 (99.0) |
| | 50 - 999 | 0 (0.0) | 0 (0.0) | 0 (0.0) |
| | >= 1000 | 1 (1.0) | 0 (0.0) | 1 (0.5) |
| | (Missing) | 1 (1.0) | 0 (0.0) | 1 (0.5) |
| Enrolled before or after guideline change to allow CCMDD script renewal on same day as viral load test | Before | 52 (52.0) | 51 (51.0) | 103 (51.5) |
| | After | 48 (48.0) | 49 (49.0) | 97 (48.5) |
| **Social information** | | | | |
| Hazardous drinking (AUDIT-C) | No | 76 (76.0) | 66 (66.0) | 142 (71.0) |
| | Yes | 24 (24.0) | 34 (34.0) | 58 (29.0) |
| Highest level of education completed | None | 2 (2.0) | 3 (3.0) | 5 (2.5) |
| | Primary school only | 13 (13.0) | 5 (5.0) | 18 (9.0) |
| | Secondary school but not matric | 32 (32.0) | 43 (43.0) | 75 (37.5) |
| | Matriculation | 46 (46.0) | 39 (39.0) | 85 (42.5) |
| | Tertiary | 7 (7.0) | 10 (10.0) | 17 (8.5) |
| Employment status | Unemployed | 29 (29.0) | 20 (20.0) | 49 (24.5) |
| | Informal employment | 4 (4.0) | 2 (2.0) | 6 (3.0) |
| | Part time employment | 18 (18.0) | 24 (24.0) | 42 (21.0) |
| | Full-time employment | 41 (41.0) | 44 (44.0) | 85 (42.5) |
| | Student | 1 (1.0) | 1 (1.0) | 2 (1.0) |
| | Self-employed | 7 (7.0) | 9 (9.0) | 16 (8.0) |
| Monthly personal income, ZAR | Median (IQR) | 3000.0 (0.0 to 5000.0) | 4000.0 (1200.0 to 6000.0) | 3500.0 (350.0 to 5000.0) |

*(Continued)*

**Table 1.** (Continued)

| Variable | Levels | Intervention arm | Standard-of-care arm | Total |
|---|---|---|---|---|
| Mode of transport to clinic | Private transport | 5 (5.0) | 5 (5.0) | 10 (5.0) |
| | Public transport | 89 (89.0) | 92 (92.0) | 181 (90.5) |
| | Walking | 6 (6.0) | 3 (3.0) | 9 (4.5) |
| Travel time to clinic and back, minutes | Median (IQR) | 60.0 (47.5 to 90.0) | 60.0 (50.0 to 82.5) | 60.0 (50.0 to 90.0) |
| Cost of travel to clinic and back, ZAR | Median (IQR) | 40.0 (26.0 to 50.0) | 45.0 (30.0 to 50.0) | 44.0 (28.0 to 50.0) |
| Mode of transport to PuP | Private transport | 6 (6.0) | 4 (4.0) | 10 (5.0) |
| | Public transport | 82 (82.0) | 83 (83.0) | 165 (82.5) |
| | Walking | 12 (12.0) | 13 (13.0) | 25 (12.5) |
| Travel time to PuP and back, minutes | Median (IQR) | 50.0 (30.0 to 60.0) | 50.0 (30.0 to 62.5) | 50.0 (30.0 to 60.0) |
| Cost of travel to PuP and back, ZAR | Median (IQR) | 30.0 (20.0 to 50.0) | 30.0 (20.0 to 50.0) | 30.0 (20.0 to 50.0) |

**Table 2. PHILA study enrolment visit viral loads and CCMDD referral data.**

| Variable | Levels | Intervention arm | Standard-of-care arm | Total |
|---|---|---|---|---|
| Enrolment viral load (cat) | < 50 | 94 (94.0) | 92 (92.0) | 186 (93.0) |
| | 50 - 999 | 2 (2.0) | 3 (3.0) | 5 (2.5) |
| | >= 1000 | 0 (0.0) | 2 (2.0) | 2 (1.0) |
| | Missing | 4* (4.0) | 3** (3.0) | 7 (3.5) |
| Enrolment creatinine, µmol/L | Median (IQR) | 82.5 (75.0 to 97.0) | 83.5 (72.0 to 91.2) | 82.5 (73.0 to 93.2) |
| CCMDD pick-up point type | Pharmacy | 68 (68.0) | 71 (71.0) | 139 (69.5) |
| | Pele-Box*** | 27 (27.0) | 20 (20.0) | 47 (23.5) |
| | Other | 2 (2.0) | 3 (3.0) | 5 (2.5) |
| | Not referred | 3 (3.0) | 6 (6.0) | 9 (4.5) |
| Number of ART cycles prescribed | 2 | 78 (78.0) | 82 (82.0) | 160 (80.0) |
| | 3 | 19 (19.0) | 12 (12.0) | 31 (15.5) |
| | Not referred to CCMDD | 3 (3.0) | 6 (6.0) | 9 (4.5) |
| Months of ART per cycle | 2 | 19 (19.0) | 12 (12.0) | 31 (15.5) |
| | 3 | 78 (78.0) | 82 (82.0) | 160 (80.0) |
| | Not referred to CCMDD | 3 (3.0) | 6 (6.0) | 9 (4.5) |

\* 4 invalid point-of-care enrolment viral load results repeated on the same day and <50 copies/mL.\* 3 invalid laboratory enrolment viral load results repeated at next visit and <50 copies/mL.\*\*\*Automated smart locker [12].

## Primary outcome

In the intervention arm, 93/100 (93.0%) of participants had a CCMDD prescription renewal within three weeks, compared to 81/100 (81.0%) in the standard-of-care arm (absolute risk difference [RD] 12.0%, 95% confidence interval [CI] 2.9 to 21.2%, p = 0.021, Table 3). In a post-hoc sensitivity analysis, the point estimate of the effect of the intervention was greater prior to the guideline change to allow same day CCMDD prescription renewal (RD 23.6%, 95% CI 10.3 to 36.9), than after (RD 0.2%, 95% CI -12.3 to 11.9), although the likelihood ratio test for an interaction was not significant (p = 0.351).

## Secondary outcomes

Participants received their enrolment VL results after a median of 0 days (IQR 0–0) in the intervention arm and 20 days (IQR 7 to not received) in the standard-of-care arm (Fig B in S1 Text), with more participants receiving their results in the intervention versus standard-of-care arm at all timepoints through to the end of follow-up at 16 weeks. Of note, after the

**Global Public Health** PLOS

**Table 3. Primary and secondary outcomes for the PHILA study, n = 200.**

| | Intervention n/N (%) | Standard care n/N (%) | Absolute risk difference in % points (95% CI) | p value |
|---|---|---|---|---|
| **Primary outcome** | | | | |
| CCMDD prescription renewal within 3 weeks | 93/100 (93.0%) | 81/100 (81.0%) | 12.0% (2.9 to 21.2) | 0.021 |
| Before guideline change* | 50/52 (96.2%) | 37/51 (72.5%) | 23.6% (10.3 to 36.9) | 0.351** |
| After guideline change* | 43/48 (89.6%) | 44/49 (89.8%) | -0.2% (-12.3 to 11.9) | |
| **Secondary outcomes** | | | | |
| Time to participant receiving enrolment VL result, days (median, IQR) | 0 (0, 0) | 20 (7, Not received) | – | – |
| Participants who received viral load results within: | | | | |
| 2 weeks | 100/100 (100%) | 42/100 (42%) | 58% [47.33 - 68.67] | <0.001 |
| 4 weeks | 100/100 (100%) | 59/100 (59%) | 41% [30.36 - 51.64] | <0.001 |
| 8 weeks | 100/100 (100%) | 63/100 (63%) | 37% [26.54 - 47.46] | <0.001 |
| 12 weeks | 100/100 (100%) | 65/100 (65%) | 35% [24.65 - 45.35] | <0.001 |
| 16 weeks | 100/100 (100%) | 65/100 (65%) | 35% [24.65 - 45.35] | <0.001 |
| Time to CCMDD prescription renewal, days (median, IQR) | 0 (0, 0) | 7 (0, 12) | – | – |
| CCMDD prescription renewal within: | | | | |
| 2 weeks | 93/100 (93%) | 76/100 (76%) | 17% [6.25 - 27.75] | 0.002 |
| 4 weeks | 93/100 (93%) | 88/100 (88%) | 5% [-4.1 - 14.1] | 0.335 |
| 8 weeks | 95/100 (95%) | 92/100 (92%) | 3% [-4.82 - 10.82] | 0.566 |
| 12 weeks | 95/100 (95%) | 92/100 (92%) | 3% [-4.82 - 10.82] | 0.566 |
| 16 weeks | 97/100 (97%) | 94/100 (94%) | 3% [-3.73 - 9.73] | 0.495 |
| Time to first CCMDD ART collection, days (median, IQR) | 84 (83–89) | 91 (84–105) | – | – |
| First CCMDD ART collection within: | | | | |
| 2 weeks | 0/100 (0.0%) | 0/100 (0%) | 0% [0 - 0] | NA |
| 4 weeks | 0/100 (0.0%) | 0/100 (0%) | 0% [0 - 0] | NA |
| 8 weeks | 3/100 (3.0%) | 2/100 (2%) | 1% [-4.33 - 6.33] | 1.000 |
| 12 weeks | 29/100 (29.0%) | 15/100 (15%) | 14% [1.68 - 26.32] | 0.026 |
| 16 weeks | 80/100 (80.0%) | 78/100 (78%) | 2.0% [-10.29 - 14.29] | 0.862 |
| Proportion of participants retained-in-care | 89/100 (89.0%) | 87/100 (87%) | 2.0% (-8.0 to 12.0) | 0.828 |
| | **Mean (95% CI)** | **Mean (95% CI)** | **Mean difference (95% CI)** | |
| Number of clinic visits required for CCMDD renewal | 1.06 (1.00 - 1.12) | 1.60 (1.47 -1.73) | -0.54 (-0.40 to -0.68) | <0.001 |
| Travel costs from the patient perspective to have CCMDD prescription renewed (ZAR) | 47.7 (39.6 – 55.8) | 72.8 (59.0 – 86.7) | -25.1 (-9.2 to -41.1) | <0.001 |

* On May 24th, 2023 new South African guidelines were implemented which recommended that in the standard-of-care arm, clients could have their CCMDD prescription renewed on the same day as their VL was taken, without waiting for the results.

**p for interaction using likelihood ratio test.

guideline change, 34/49 (69.4%) in the standard-of-care arm had their CCMDD prescription renewed on the same day that their enrolment laboratory VL blood was drawn. Of these, 32/34 (94.1%) were <50 copies/mL and 30/32 (93.8%) reached the end of follow up without receiving their VL result (Table A in S1 Text). 1/34 (2.9%) had an invalid enrolment laboratory VL and was called back for a repeat test after 5 days, and 1/34 (2.9%) had a VL of 120 copies/mL, but had no further clinic visits before the end of follow-up.

The median time to CCMDD prescription renewal was 0 days in the intervention arm (IQR 0–0) and 7 days in the standard-of-care arm (IQR 0–8; Fig 2A). The proportion with CCMDD renewal by 2 weeks was higher in the intervention arm (93/100 (93.0%) versus 76/100 (76.0%), RD 17.0% (6.3 – 27.8) p = 0.001), but was not significantly different by 4

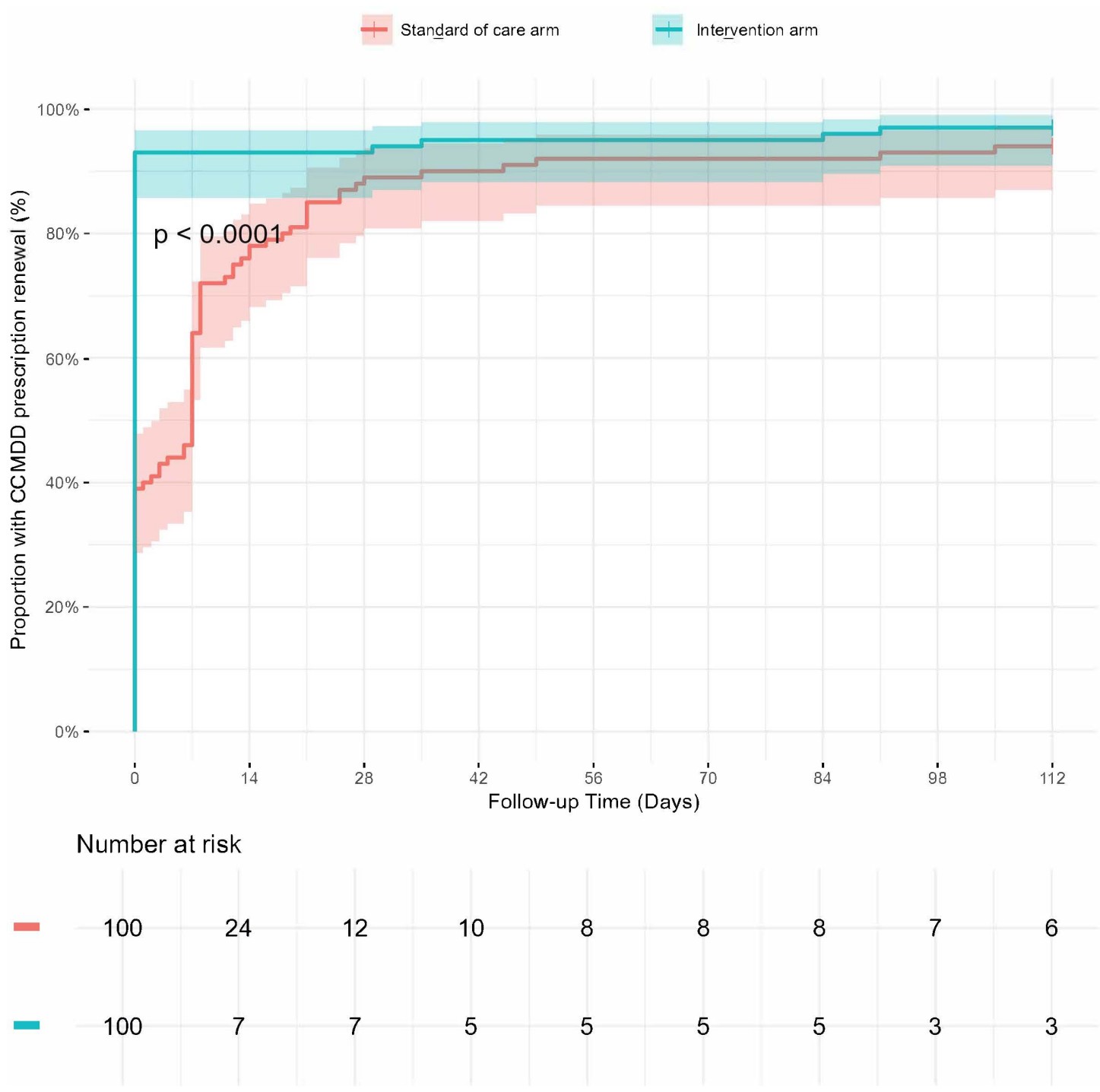

**Fig 2. Time to CCMDD prescription renewal in the PHILA trial.**

weeks onwards (Table 3). By the end of follow-up, 97% in the intervention arm and 94% in the standard-of-care arm had a CCMDD prescription renewal. The most common type of pickup points that participants were referred to were private pharmacies (n = 139), followed by 'Pele Boxes' (n = 47, Table 2). The majority (n = 160, 83.8%) were prescribed two cycles of three months of ART, as opposed to three cycles of two months ART supply (n = 31, 16.2%), which was more common early in the trial (Fig C in S1 Text).

The median time to first CCMDD ART collection was 84 days (IQR 83–89) in the intervention arm and 91 (IQR 84–105) in the standard-of-care arm (Fig D in S1 Text). The proportion with CCMDD ART collection at 12 weeks was higher in the intervention arm (29.0%) than in the standard-of-care arm (15%, RD 14.0%, 95% CI 1.7 to 26.3), but not different at any other timepoint. There was no difference between arms in the proportion retained-in-care between 8 and 16 weeks (89.0% versus 87.0%, RD 2.0% 95% CI -8.0 to 12.0), and the main reasons for not collecting ART are listed in Table B in S1 Text. The mean number of clinic visits required for CCMDD renewal was lower in the intervention arm (1.06) versus the standard-of-care arm (1.60, mean difference -0.54, 95% CI -0.40 to -0.68), as was the total travel cost to participants to have their CCMDD prescription renewed (ZAR 47.7 versus ZAR 72.8, mean difference ZAR -25.1 [95% CI -9.2 to -41.1]).

During follow-up, there was one hospitalisation, resulting in death due to pneumonia in the intervention arm. This was not deemed to be related to the intervention.

### Outcomes of CCMDD referral and laboratory VL testing in KwaZulu-Natal clinics

Between 24th May 2023 and 10th September 2023 there were 16,568 adults in CCMDD with a CCMDD prescription renewal on the same day as blood draw in 108 public KwaZulu-Natal clinics (Table C in S1 Text). Of these, 2,932/16,568 (17.7%) were >50 copies/mL (with 373/16,568, 2.3% ≥ 1000 copies/mL). Of those with a viral load >50 copies/mL, 25/2932 (0.9%) had a clinical visit with a nurse/doctor within 16 weeks, at a median of 88 days (IQR 84–101), and 111/2,932 (3.8%) had a repeat VL taken within 16 weeks, after a median of 86 (IQR 75.5-94.5) days. Of those with a viral load ≥1000 copies/mL, 2/373 (0.5%) had a clinical visit within 16 weeks (after 96 and 107 days) and 21/373 (5.6%) had a repeat VL within 16 weeks (after a median of 84 (IQR 68–92) days).

## Discussion

### Summary

In this randomised controlled trial we found that point-of-care testing reduced the number of people with a delayed CCMDD prescription renewal of three or more weeks, who are defined as 'dormant clients' in the CCMDD programme, but did not improve overall retention-in-care or time to CCMDD ART collection. Point-of-care testing also reduced the time to receipt of VL results, and the number of clinic visits, and associated travel costs, required by participants for CCMDD prescription renewal.

### Interpretation and comparison with other studies

These results suggest that point-of-care testing is a potential strategy to improve efficiency in the CCMDD programme, and other community ART delivery programmes, by reducing the number of clinic visits (and associated travel costs) required to confirm viral suppression and CCMDD eligibility. However, the change in South African guidelines to allow CCMDD prescription renewal without review of laboratory results also led to quicker renewals and a reduction in clinic visits in the standard-of-care arm, but with the effect of greatly increasing the time to patients receiving their VL results. It is important that people know their VL results to increase self-efficacy and self-management, which has been associated with better engagement in care [21]. In particular, given that people with an undetectable VL cannot transmit HIV (promoted in the undetectable = untransmittable [U = U] campaign), knowledge of VL may guide sexual behaviour and is important in preventing HIV transmission [22,23]. The one participant in PHILA who was referred into CCMDD and subsequently turned out to have an unsuppressed VL was not seen again during follow-up. And in our post-hoc analysis

in 108 public sector clinics during the early stages of the guideline change, the vast majority of people with viraemia who had their CCMDD prescriptions on the same day, had no follow-up clinic visits or repeat VL within 16 weeks. This shows that systems to review results when clients are not in the clinic need to be improved. Unless these people were phoned (which is not captured in the routine data system) they could have been viraemic, without knowing, until their next clinic visit after 6 months. Our analysis did not take into account all people in CCMDD who were due a viral load, nor compare before and after the guidelines within routine care, and so we plan a separate, more complete analysis with extended follow-up data, to better assess the overall impact of this guideline change and whether it improved CCMDD prescription renewal, and its impact on management of viraemia in CCMDD in routine care. Models of differentiated service delivery aim to increase efficiency in the healthcare system and also better serve the needs of PLWH, and so balancing the need for efficient CCMDD prescription renewal with provision of results is important. Point-of-care VL testing may help achieve both aims, but additional strategies to provide all VL results, without clients needing to attend clinic should be explored.

Evidence around the impact of point-of-care VL testing varies depending on the type of study. Similar to our findings, several randomised trials have shown the point-of-care VL testing can improve availability of VL results, but none that we are aware of have assessed the impact in a differentiated ART delivery programmes such as those used in CCMDD [9]. A randomised controlled trial in adults with HIV who had recently initiated ART found point-of-care VL testing was associated with better 12-month viral suppression and retention-in-care [24], but other trials in adults [25], children [26], pregnant women [27], adolescents [28] and people with viraemia [29], have not found an impact on viral suppression and retention-in-care. We did not assess viral suppression, but found no effect on short term retention-in-care. Studies in more routine settings have found more rapid availability of results, but results have not always led to rapid clinical action or improved outcomes [30–32]. Qualitative research suggests that clients appreciate quicker, same-day VL results, which may facilitate better adherence, but there have been concerns around feasibility of implementing point-of-care VL testing in routine healthcare settings [33,34].

## Strengths and limitations

Strengths of our study include the randomized design, the high fidelity to the point-of-care VL intervention and the use of routine healthcare staff and settings, with management following South African guidelines. We did not assess a viral suppression outcome, as given the high proportion of people with viral suppression at baseline, it is unlikely that point-of-care VL could improve this outcome. Instead, we focussed on a programmatic outcome measure (dormancy) that is used by the South African Department of Health to measure efficiency in the CCMDD programme, making our results directly relevant to local policy makers. We had planned for VL testing to be conducted by nurses in the clinic, but staff shortages and changes in services from the COVID-19 pandemic meant that research site laboratory staff conducted most of the testing.

## Implications for research and policy

We show that point-of-care VL has potential to improve the efficiency of differentiated ART delivery programmes, without compromising on providing VL results to clients. In differentiated care programs where annual ART prescriptions are used [35], point-of-care VL could allow stable clients to attend clinic only once a year [9]. However, further work to determine what is required to successfully implement point-of-care VL testing in routine, non-research settings is required, and we will present qualitative work around the implementation of point-of-care VL testing in a separate manuscript. Health economic studies that account for costs of point-of-care viral load implementation, and potential healthcare system and client cost savings, are also required. Furthermore, better VL result review processes and call back of people with a high VL are required within the current policy of allowing CCMDD renewal on the same day as laboratory VL testing. Alternative strategies to provide VL results remotely, using phone calls or text messaging, should also be explored. While these strategies have been used successfully in high-income settings to provide sexually transmitted infection results [36], maintaining

confidentiality in low- and middle-income country settings can be difficult due to phones often being shared [34], and so further work to develop these strategies is required [37].

## Conclusion

In this randomized controlled trial we showed that point-of-care VL testing improves the proportion of people with CCMDD ART prescription renewal within three weeks, by reducing time to availability of VL results for healthcare workers and clients, and reduces clinic visits and associated travel costs for clients.

## Supporting information

**S1 Text. Figures and tables for point-of-care HIV viral load testing in a community antiretroviral therapy delivery programme: A randomised controlled trial (PHILA).**
(DOCX)

**S1 Checklist. CONSORT checklist: Reproduced from the CONSORT statement under the creative commons attribution 4.0 international (CC BY 4.0) licence.**
(DOCX)

## Acknowledgments

The authors would like to thank all participants in the study and acknowledge the work and support of staff at the Prince Cyril Zulu Clinic, eThekwini Municipality, CAPRISA and the National Health Laboratory Services at Addington and Inkosi Albert Luthuli Hospitals.

## Author contributions

**Conceptualization:** Jienchi Dorward, Yukteshwar Sookrajh, Nigel Garrett.

**Data curation:** Jienchi Dorward, Jennifer A Brown, Lara Lewis.

**Formal analysis:** Jienchi Dorward, Jennifer A Brown, Lara Lewis.

**Funding acquisition:** Jienchi Dorward, Christopher C Butler, Gail Hayward, Nigel Garrett.

**Investigation:** Jienchi Dorward, Kwena Tlhaku, Yukteshwar Sookrajh, Pedzisai Munatsi, Jessica Naidoo, Emelda Tselana, Andile Maphumulo, Nokuthandwa Mbambo, Thobile Mhlongo-Gumbi, Pravikrishnen Moodley, Natasha Samsunder, Paul K. Drain, Christopher C Butler, Gail Hayward, Nigel Garrett.

**Methodology:** Jienchi Dorward, Yukteshwar Sookrajh, Jessica Naidoo, Emelda Tselana, Pravikrishnen Moodley, Natasha Samsunder, Paul K. Drain, Christopher C Butler, Gail Hayward, Nigel Garrett.

**Project administration:** Jienchi Dorward, Kwena Tlhaku, Pedzisai Munatsi, Jessica Naidoo, Andile Maphumulo, Christopher C Butler, Gail Hayward, Nigel Garrett.

**Resources:** Christopher C Butler, Gail Hayward, Nigel Garrett.

**Supervision:** Jienchi Dorward, Christopher C Butler, Gail Hayward, Nigel Garrett.

**Validation:** Jienchi Dorward, Andile Maphumulo.

**Writing – original draft:** Jienchi Dorward, Pravikrishnen Moodley.

**Writing – review & editing:** Jienchi Dorward, Kwena Tlhaku, Yukteshwar Sookrajh, Pedzisai Munatsi, Jessica Naidoo, Emelda Tselana, Andile Maphumulo, Nokuthandwa Mbambo, Thobile Mhlongo-Gumbi, Jennifer A Brown, Lara Lewis, Natasha Samsunder, Paul K. Drain, Christopher C Butler, Gail Hayward, Nigel Garrett.

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
