## [Decision Letter · Decision Letter 0]

20 Oct 2025

PGPH-D-25-01667

Point-of-care HIV viral load testing in a community antiretroviral therapy delivery programme: a randomised controlled trial (PHILA)

Dear Dr. Jienchi Dorward

Thank you for submitting your manuscript to PLOS Global Public Health. After careful consideration, we feel that it has merit but does not fully meet PLOS Global Public Health’s publication criteria as it currently stands. Therefore, we invite you to submit a revised version of the manuscript that addresses the points raised during the review process.

Please see the points in particular raised by Reviewer 2 and do respond to both reviewers.

We look forward to receiving your revised manuscript.

Kind regards,

Megan Coffee, MD, PhD

Academic Editor

Journal Requirements:

1. Please provide a detailed online Financial Disclosure statement. This is published with the article. It must therefore be completed in full sentences and contain the exact wording you wish to be published.

a) Please clarify all sources of financial support for your study. List the grants, grant numbers, and organizations that funded your study, including funding received from your institution. Please note that suppliers of material support, including research materials, should be recognized in the Acknowledgements section rather than in the Financial Disclosure.

b) State the initials, alongside each funding source, of each author to receive each grant. For example: “This work was supported by the National Institutes of Health (####### to AM; ###### to CJ) and the National Science Foundation (###### to AM).”

c) State what role the funders took in the study. If the funders had no role in your study, please state: “The funders had no role in study design, data collection and analysis, decision to publish, or preparation of the manuscript.”

For more information, please go to our submission guidelines:

https://journals.plos.org/globalpublichealth/s/submission-guidelines#loc-financial-disclosure-statement

2. Please ensure that the funders and grant numbers match between the Financial Disclosure field and the Funding Information tab in your submission form. Note that the funders must be provided in the same order in both places as well.

3. In the online submission form, you indicated that “Bona fide researchers will be able to request access to anonymised trial data by contacting the corresponding author.”.

a) In a public repository,

b) Within the manuscript itself, or

c) Uploaded as supplementary information.

4. Please provide separate main figure files in .tif or .eps format only and ensure that all files are under our size limit of 10MB.

5. Please ensure that you refer to Figure 1 in your text as, if accepted, production will need this reference to link the reader to the figure.

6. Please include a separate legend or caption for Figure 1 in your manuscript.

7. We notice that your supplementary figures and tables are included in the manuscript file. Please remove them and upload them with the file type 'Supporting Information'. Please ensure that each Supporting Information file has a legend listed in the manuscript before or after the references list.

8. We have noticed that you have uploaded Supporting Information files, but you have not included a list of legends. Please add a full list of legends for your Supporting Information files before or after the references list.

9. Some material included in your submission may be copyrighted. According to PLOS’s copyright policy, authors who use figures or other material (e.g., graphics, clipart, maps) from another author or copyright holder must demonstrate or obtain permission to publish this material under the Creative Commons Attribution 4.0 International (CC BY 4.0) License used by PLOS journals. Please closely review the details of PLOS’s copyright requirements here: PLOS Licenses and Copyright. If you need to request permissions from a copyright holder, you may use PLOS's Copyright Content Permission form.

Potential Copyright Issues:

‘Figure 1’ and ‘Figure 2’ in “PHILA_POC_VL_CCMDD_Protocol_Version_2.0_10Nov21_clean.pdf”: Please confirm whether you drew the images / clip-art within the figure panels by hand. If you did not draw the images, please provide (a) a link to the source of the images or icons and their license / terms of use; or (b) written permission from the copyright holder to publish the images or icons under our CC-BY 4.0 license. Alternatively, you may replace the images with open source alternatives. See these open source resources you may use to replace images / clip-art:

- https://openclipart.org/

Additional Editor Comments (if provided):

Reviewers' comments:

Reviewer's Responses to Questions

**Comments to the Author**

1. Does this manuscript meet PLOS Global Public Health’s publication criteria?

Reviewer #1: Yes

Reviewer #2: Yes

2. Has the statistical analysis been performed appropriately and rigorously?

Reviewer #1: No

Reviewer #2: Yes

3. Have the authors made all data underlying the findings in their manuscript fully available (please refer to the Data Availability Statement at the start of the manuscript PDF file)?

Reviewer #1: No

Reviewer #2: Yes

4. Is the manuscript presented in an intelligible fashion and written in standard English?

Reviewer #1: Yes

Reviewer #2: Yes

Reviewer #1: General comments:

This manuscript reports the results of an open-label, randomized controlled trial of point-of-care HIV viral load (VL) testing versus laboratory-based VL testing on antiretroviral therapy (ART) prescription renewal proportions among people living with HIV in Durban, South Africa.

I have a couple of general comments about the analytic approach for this trial. In expectation, randomization will "balance" the treatment groups of a randomized trial. As is often the case with a randomized study, many variables have similar distributions in the treatment groups, but there are a few that differ. Given your sample size, I believe you could also control for some of these covariates that differ between groups. I would encourage moving to a multivariable model to achieve this. The choice of method will depend on how you want to report your estimate.

That leads to my second general comment: I believe the risk difference quantity is in percentage points and not percentage. For example, in the abstract (line 69) a percentage difference, should be ~15% since 93/81=1.148... and not 12%. Thus, you'll either need to change to percentage differences or change the reporting and language to indicate that this is a percentage point difference.

Specific comments:

1. (line 99) Typo ("0")

2. (lines 147-149) What algorithm or software was used to random participants? How was the randomization implemented? What block sizes were used?

3. (lines 204-207) What methods were used to determine power?

4. (line 215) Please provide a methodological citation for the Newcombe-Wilson confidence interval method.

5. (line 219) Please provide a methodological citation for your regression model.

6. (Figure 2) I suggest changing the y-axis label to something more appropriate. You may also wish to invert the y-axis since the outcome is a positive, i.e., a higher probability is a good thing. (The same applies to Figures S3 and S4.)

7. (Table 1) I'd suggest including standardized mean differences in this table to allow for easier comparison of differences between the intervention and SOC arms.

Reviewer #2: Overall comments:

This paper seeks to evaluate the impact of POC VL testing in the CCMDD renewal process in South Africa. This appears to be a well-done trial with some interesting findings regarding where POC VL did and did not have impacts. Most of the comments are just focused on getting some additional clarity on expected workflows/processes to help contextualize and interpret the varied results (and why some changed while others didn’t).

Intro

- May be helpful to contextualize VLs as a health system barrier that leads to unnecessary “dormancy”. Might also be worth briefly identifying the other reasons (e.g., client-related, treatment interruptions) or any other health system provider level.

- I think could be worth being clear that this is also a self-inflicted problem, specifically with CCMDD continuation. It is not obvious or necessary that the health system would decide to sort of pause CCMDD until VL can be confirmed. But it is also a fact that the SA guidelines decided to do this (at least for the first part of the study). But just clarifying that specific of the SA guideline that people are expected to confirm the VL results before allowing clients to continue provides some straightforward context because it it not intuitive that this challenge would be addressed this way (I would have imagined the problem would more so be people receive CCMDD, and then poor follow up for those who end up having elevated VLs).

Methods

- Under Trial (or Control description), would state what happens regarding ART/CCMDD once VL drawn. Are clients given a one month ART supply from clinic, told to come back to clinic where VL is reviewed, and then referred back to CCMDD? Also, how does that relate to CCMDD dormany if the VL takes 3 weeks to come back? Are they then classified as dormant automatically?

- With regards to outcomes, I wasn’t completely clear on the distinction between CCMDD referall, prescription, ART pick up. My sense is that the clinic writes a referall/prescription and then clients later go and pick up their medications from CCMDD. I also wasn’t sure how this relates to the ART supply a client may have on hand (e.g., some places align the provided drug supply with the next pharmacy appointment) and what all the different point where client becomes at risk for having a lapse in ART.

Results

- I might try to report the secondary outcomes essentially in cascade order. Time to VL results, time to prescription, time to ART pick up.

- The fact that so few people had CCMDD pick up by 12 weeks made me question if I understood the standard procedures well. Is it that at the time of VL check people are still given 2 to 3 months of medicine…so the main concern with POC vs. SOC VL measurements is the time to getting a result and the need for second visit, rather than a real risk of ART lapse. I initially interpreted CCMDD dormancy as akin to a treatment interruption but perhaps that is not the case. Hence, the comments for clarity in the methods (I do now see some/much of this is clearer with the Figure in the supplement…but would be good to put in main text as the understanding workflow key for interpreting results).

- Do SOC procedures make it normal for a CCMDD pick up to be later? E.g., receive 1 month until VL results come back, and then 3 months at the VL check visit…so you don’t go to CCMDD again for 4 months (vs. maybe 3 months if you had POC VL).

- Would be nice to see all secondary outcomes by before and after guideline change in supplement.

- I wasn’t clear why there would be no differences in CCMDD prescriptions but a difference in actually receiving VL (as one should be contingent on other). Althought there weren’t big gaps in CCMDD, there does seem to be a big gap in the receipt of VL results, and unclear what this means. There is likely relevance even if it doesn’t end up impacting CCMDD pathway.

- Might be interesting to understand the differences in what happened among those who had elevated VLs (e.g., time to EAC, suppression, back to CCMDD). The numbers are likely quite small but could warrant a reference as this is somewhere else I might imagine POC VL could have impacts.

- The larger analysis with Tier data is clearly related, but seems a little out of place. It seems like it could stand on its own as a paper (and the trial could easily as well). I think it might fit better or be more informative if the exact same type of analyses were performed in this wider dataset before and after the guideline change. In that way, it sort of serves as an alternative control arm that may not be effected by Hawthorne effect of being in a trial.

- Even without this, I think would be important to see the before and after guideline change results (rather than just the after). Additionally, I think you would want to include outcomes among everyone in CCMDD that was due for VL rather than also restrict to those who also got CCMDD prescription. I think this would give a broader picture of the guideline change (perhaps more appropriate people in CCMDD, but also more inappropriate…other gaps will probably become evident).

Discussion

- For summary, I would also include the important “null” results. My main takeaway honestly was that people did similarly well for the outcomes we probably care about most (i.e., potential lapse in ART), but there are still other operational and client-associated benefits with regards to receiving the VL results, not having to make additional visit. Although 3 week outcome was selected as primary (and there is some obligation to focus on it in reporting), I don’t think it really provides the appropriate overall story and would try to balance those tensions (For me, articulating the clearest narrative is most important even if it sometimes at odds with the “rules” of clinical trial reporting). That being said, the rest of the discussion provides an excellent narrative synopsis of the meaning of the totality of results.

- Is there anything to be said regarding cost of implementing vs. cost (and time) savings from client end?

Aaloke Mody

**Do you want your identity to be public for this peer review?** For information about this choice, including consent withdrawal, please see our Privacy Policy

Reviewer #1: No

Reviewer #2: **Yes:** Aaloke Mody

---

## [Decision Letter · Decision Letter 1]

12 Jan 2026

Point-of-care HIV viral load testing in a community antiretroviral therapy delivery programme: a randomised controlled trial (PHILA)

PGPH-D-25-01667R1

Dear Dr Dorward

We are pleased to inform you that your manuscript 'Point-of-care HIV viral load testing in a community antiretroviral therapy delivery programme: a randomised controlled trial (PHILA)' has been provisionally accepted for publication in PLOS Global Public Health.

Best regards,

Megan Coffee, MD, PhD

Academic Editor

Reviewer Comments (if any, and for reference):

Reviewer's Responses to Questions

**Comments to the Author**

Reviewer #1: All comments have been addressed

Reviewer #2: All comments have been addressed

publication criteria?

Reviewer #1: (No Response)

Reviewer #2: Yes

3. Has the statistical analysis been performed appropriately and rigorously?

Reviewer #1: (No Response)

Reviewer #2: Yes

4. Have the authors made all data underlying the findings in their manuscript fully available (please refer to the Data Availability Statement at the start of the manuscript PDF file)?

Reviewer #1: (No Response)

Reviewer #2: Yes

5. Is the manuscript presented in an intelligible fashion and written in standard English?

Reviewer #1: (No Response)

Reviewer #2: Yes

Reviewer #1: (No Response)

Reviewer #2: Thank you for addresing all of my prior comments.

**Do you want your identity to be public for this peer review?** For information about this choice, including consent withdrawal, please see our Privacy Policy

Reviewer #1: No

Reviewer #2: **Yes:** Aaloke Mody
